# The Role of Oxidative Stress in Autism Spectrum Disorder Pathophysiology, Diagnosis and Treatment

**DOI:** 10.3390/biomedicines13020388

**Published:** 2025-02-06

**Authors:** Aleksandra Kuźniar-Pałka

**Affiliations:** Clinic of Pediatric and Adolescent Neurology, Institute of Mother and Child, 01-211 Warsaw, Poland; alekuzniar@imid.med.pl

**Keywords:** ASD, oxidative stress, oxidative stress biomarkers, antioxidants

## Abstract

Autism spectrum disorder (ASD) is a significant health problem with no known single cause. There is a vast number of evidence to suggest that oxidative stress plays an important role in this disorder. The author of this article reviewed the current literature in order to summarise the knowledge on the subject. In this paper, the role of oxidative stress is investigated in the context of its influence on pathogenesis, the use of oxidative stress biomarkers as diagnostic tools and the use of antioxidants in ASD treatment. Given the heterogeneity of ASD aetiology and inadequate treatment approaches, the search for common metabolic traits is essential to find more efficient diagnostic tools and treatment methods. There are increasing data to suggest that oxidative stress is involved in the pathogenesis of ASD, both directly and through its interplay with inflammation and mitochondrial dysfunction. Oxidative stress biomarkers appear to have good potential to be used as diagnostic tools to aid early diagnosis of ASD. The results are most promising for glutathione and its derivatives and also for isoprostanses. Probably, complex dedicated multi-parametric metabolic panels may be used in the future. Antioxidants show good potential in ASD-supportive treatment. In all described fields, the data support the importance of oxidative stress but also a need for further research, especially in the context of sample size and, preferably, with a multicentre approach.

## 1. Introduction

Autism spectrum disorder (ASD) is a global problem, sometimes called a hidden disability. This is a neurodevelopmental disease described according to the DSM V (Diagnostic and Statistical Manual of Mental Disorders 5th edition) criteria as a type of pervasive developmental disorder that is defined by (a) persistent deficits in social communication and social interaction across multiple contexts; (b) restricted, repetitive patterns of behaviour, interests or activities; and (c) symptoms that are present in the early developmental period and cause impairment in multiple areas of functioning. The aetiology of this disorder is multifactorial. The prevalence of ASD in children has increased over the last three decades. The median prevalence of ASD in children is 1/100, with a range of 1.09/10,000 to 436.0/10,000 [1,2]. The data show that autism spectrum disorder (ASD) is diagnosed more often in males than in females. The male-to-female ratio is usually reported to be around 4:1 [2]. The diagnosis is made by evaluating the patient according to the behavioural and clinical criteria described in the DSM V or the 10th revision of the International Classification of Diseases (ICD 10).

The average age of diagnosis is 3.5 years [3], although some symptoms are usually apparent from the first year of life. There is currently no cure for ASD. The most effective interventions are behavioural and environmental therapy accompanied by medication to relieve symptoms. Early diagnosis would allow for appropriate treatment to begin earlier. Earlier therapeutic intervention may lead to better outcomes for people with ASD [4]. To date, there is no single biomarker, neuroimaging finding or genetic mutation that is sufficient enough for accurate ASD diagnosis [5]. This is due to the absence of a single, predominant factor that has been identified as a causative agent for autism. Nowadays, there is increasing evidence of the important multimodal role of oxidative stress in ASD. The aim of this article is to explore how the knowledge of oxidative stress in ASD can be used in clinical practice. There is considerable evidence for the presence of oxidative stress in individuals with ASD [5,6,7,8,9,10,11,12,13] and some also in their parents [10]. The evidence is derived from metabolic studies conducted on patients diagnosed with ASD, and a more comprehensive discussion of this evidence can be found in the section on the role of oxidative stress in the pathophysiology of ASD.

Oxidative stress is described as an imbalance between reactive oxygen species (ROS) and reactive nitrogen species (RNS) and endogenous antioxidants. The function of antioxidant mechanisms is to balance and neutralise the effects of ROS and RNS [14]. ROS are mainly produced by mitochondria in both physiological and pathological conditions, that is, O_2_^−^ can be formed by cellular respiration, lipoxygenases (LOX) and cyclooxygenases (COX) during arachidonic acid metabolism and by endothelial and inflammatory cells [14]. The excessive concentrations of ROS and RNS can cause cell damage, influence energy metabolism and may have some impact on gene expression [15]. The brain is highly susceptible to oxidative stress due to its high energy consumption, during which a large amount of oxygen is utilised, lipid-rich content with high proportions of polyunsaturated fatty acids (PUFAs) and relatively low activity of antioxidant defence mechanisms [16]. Considering these factors, there is growing evidence that oxidative stress plays an important role in ASD and other neuropsychiatric disorders.

The following sections of this article will review the role of oxidative stress in ASD in the context of pathophysiology, its utility as an ASD biomarker and its potential treatment target.

The author conducted a comprehensive search of the PubMed, Science Direct and Google Scholar databases, with the objective of identifying contemporary articles on the subject of oxidative stress in the context of ASD written in English. The cited articles were published between 2000 and 2024. The search strategy yielded a range of publications using various combinations of search terms, including “oxidative stress and ASD”, “oxidative stress and autism”, “oxidative stress biomarkers and ASD”, “oxidative stress biomarkers and Autism”, “antioxidants and autism” and “antioxidant and ASD”. The selection of articles for inclusion in the narrative review was guided by a set of inclusion criteria, including original articles, reviews, book chapters and editorials. Conversely, articles that met the exclusion criteria, such as conference abstracts and duplicates, were excluded from the review. A total of 85% of the included papers were published within the last decade. The author’s objective was to summarise and synthesise the articles on oxidative stress, as the primary focus was to present the current state of knowledge regarding the role of oxidative stress in ASD in the context of clinical practice. In this regard, data on biomarkers and potential therapeutic substances are derived from human studies. The analysis was conducted with three primary aims: to enhance understanding of the pathophysiology of ASD, to identify future diagnostic tools and treatment targets and to explore the potential for developing new interventions.

## 2. Role of Oxidative Stress in ASD Pathophysiology

Oxidative stress is an imbalance of ROS/RNS and the antioxidant protective system, causing multimodal pathogenic effects on numerous aspects of an organism’s metabolic pathways and functions. Considering the importance of oxidative stress in that process, there are growing data on its causative function in ASD development.

In the context of autism, the concept of oxidative stress cannot be regarded as a standalone problem. Its presence in individuals diagnosed with ASD is attributable to a multitude of factors, with mitochondrial dysfunction and neuroinflammation playing a pivotal role. Mitochondria, as the primary source of cellular energy, are susceptible to oxidative attack. Consequently, oxidative stress can be considered both a cause and a downstream effect of mitochondrial dysfunction. It is important to note that oxidative stress increases during inflammation and is also induced by its toxicity on cells, which can also lead to inflammation. Due to its high energy consumption, the brain is particularly vulnerable to oxidative stress and neuroinflammation. These processes interact in a vicious cycle, as previously mentioned. These mechanisms are described in more detail in the following section.

In ASD, several genetic and environmental factors interact together, promoting excessive ROS production, decreased antioxidant capacity and mitochondrial dysfunction [17]. These abnormalities happening in the prenatal and perinatal periods promote oxidative stress that can influence epigenetic dysregulation, neuro-inflammation, cerebral injury and neuronal dysfunction, which finally leads to ASD [9,17,18]. The most recognised genetic factors are interactions between GSTT1 and GSTP1 allele variants [19] and some specific gene polymorphisms [20]. Environmental factors promoting oxidative stress are maternal neurotoxin exposure, prematurity, neonatal jaundice and use of medication during pregnancy [17,21].

As mentioned above, it is widely described that oxidative stress influences many metabolic pathways. It affects cellular membranes, DNA strand breaking, protein synthesis and amino acid side chain formation, post-translational changes and neurotransmission [9]. The cellular membrane is damaged due to lipid peroxidation.

Lipids are the most susceptible to oxidative processes, especially polyunsaturated fatty acids (PUFA). Numerous neurological studies underlined the important role of lipid peroxidation in the pathophysiology of autism spectrum disorder. The carbon–carbon double strands of PUFA are particularly prone to oxidative attack. These perturbations in membrane structure cause changes in membrane fluidity and permeability, thereby adversely affecting membrane function. Peroxidation of PUFA leads to the formation of isoprostanes and generates reactive aldehydes, such as 4-hydroxynonenal and malondialdehyde (MDA). The 4-hydroxynonenal binds to proteins and interferes with their function. ROS, by oxidising amino acids, cause the unfolding and misfolding of protein chains, leading to their malfunction or inactivity [9]. Impaired metabolic pathways further increase the level of oxidative stress.

In addition, ROS and RNS react with nucleic acids, causing DNA strand breaking, DNA–protein crosslinking and modification of purine- and pyridine-base structures, resulting in DNA mutations. Oxidation of RNA bases leads to the breakage of the nucleotide strand and causes ribosomal dysfunction [22]. The best-described oxidative stress-mediated post-translational modifications of proteins are 3-nitrotyrosine (3NT) and protein carbonyl formation that may alter protein function [23]. As a result, the above-mentioned processes interfere with protein synthesis and make further cell damage possible.

Another important aspect that needs to be highlighted in relation to the influence of oxidative stress on ASD pathophysiology is oxidative stress in the brain. A plethora of researchers describe numerous factors that influence brain metabolism, causing its increased susceptibility to oxidative stress, and highlight specific abnormalities observed in ASD patients. These include the high energy requirements of the brain, the high concentration of polyunsaturated fatty acids (PUFA) in neuronal cell membranes and the high ratio of membrane to cytoplasmic volume. Another factor is the excitotoxic nature of one of the neurotransmitters—glutamate—whose changes have been described in ASD patients [24]. The next factor is the increased oxidation of neurotransmitters, which promotes ROS, and quinones, which reduces glutathione levels. In addition, excessive ROS directly downregulate tight junction proteins and indirectly activate matrix metalloproteinases (MMPs) that contribute to the unsealing of the blood–brain barrier (BBB). Other studies show a higher activation of microglia in ASD patients, leading to greater ROS and cytokine production in a vicious cycle. To all of the above, the next highlighted factors are disturbances in mitochondrial function that impair energy production and generate excessive amounts of ROS [5,25].

The above processes are observed in patients with ASD due to their increased susceptibility to oxidative stress as a result of the multifactorial disorders seen in this population [6]. Researchers have observed reduced antioxidant defences, particularly low levels of catalase, glutathione peroxidase and vitamin E [9]. It is still debated whether oxidative stress is a cause or an effect of ASD. Below, the author attempts to summarise the factors associated with the presence of oxidative stress in ASD patients.

One of the most recognised factors causing oxidative stress in ASD is impaired antioxidant function. Among these is the abnormal metabolism of glutathione (GSH), which plays a very important role in the pathophysiology of ASD. GSH is one of the most abundant antioxidants in the body. It also plays a regulatory role on glutamate receptors, and so its disruption may be linked to neural dysfunction [15,25]. Most researchers reported a decreased glutathione to glutathione disulfide (GSH/GSSG) ratio and decreased GSH levels in the brain, lymphoblastoid cells and blood of ASD children. In the brain, a reduced glutathione redox ratio was found in the temporal cortex and cerebellum of autistic patients [8,25,26]. Additionally, widely described disturbances in antioxidant defence in ASD are related to the metabolism of sulfur amino acids [27]. Low methionine levels influence reduced protein synthesis. Low cysteine levels in ASD patients result in decreased glutathione production in this disorder. The imbalances in sulfur amino acid metabolism impair methylation processes that further influence many metabolic pathways, including mitochondrial disturbances [28]. Another antioxidant impairment described in ASD patients is increased heme oxygenase-1 reported in the parietal and frontal lobes and the cerebellum [29].

There are also variations in superoxide dismutase (SOD) activity described, especially in plasma and erythrocytes, but the data are not conclusive, probably age-dependent [30]. Although SOD is one of the most important antioxidant enzymes, comprehensive data on its activity in the brains of ASD patients are missing. There are attempts to measure SOD activity in human brain organoids that show no difference between ASD and healthy organoids [31].

Another important and widely described aspect of ASD that plays an important role in oxidative stress is inflammation [11]. There is still a debate on whether high ROS levels may cause inflammatory conditions or whether inflammation may induce oxidative stress. It is also a question of whether oxidative stress injury is a cause or a downstream effect of psychiatric disorders. Numerous research studies show the importance of the link between neuroinflammation and oxidative stress. It is proven that prevalent oxidative stress supports chronic inflammation [32,33]. High concentrations of ROS can activate signalling pathways and induce high secretion of pro-inflammatory cytokines and chemokines, which promotes further ROS production in a vicious circle [10].

In ASD, impairments associated with the innate and adaptive immune systems are described. These disturbances lead to immune system dysregulation and support the onset of pro-inflammatory conditions. These conditions may lead to oxidative stress and promote chronic inflammation in the next step. Recently, Gevezova et al. showed in their study that elevated TLR4/NOX2 signalling in B cells of ASD subjects could produce systemic oxidative inflammation, which may impact neuronal functions [34]. That is observed in addition to possibly altered metabolism in several immune cell types, particularly brain cells [8]. Specifically, researchers identified abnormal alterations in microglial and astrocyte cell activation and atypical pro-inflammatory cytokine production [34,35], immune-related gene expression and other inflammatory biomarkers [36] that occur in the central neural system (CNS) [35,37]. Above all, those alterations may contribute to the pathophysiology of ASD.

Except for direct interest in the nervous system and considering the role of inflammation, there is rising interest in the role of gastrointestinal tract disturbances and gut microbiota in many diseases including ASD [38,39]. Enteric bacteria interact directly with the intestinal epithelium and, in normal conditions, that interaction helps to maintain the integrity of the mucosal barrier, which has a protective function [40,41]. Additionally, it is important to note that some microbiota can produce neuroactive substances, such as GABA and 5-HT, and induce cytokine production. The next important fact is that fermentation of dietary fibres and resistant starch produces short-chain fatty acids (SCFAs), especially butyrate, propionate and acetate. Accumulation of propionate may influence neuroinflammation and cause gut disturbances [38]. There is rising evidence that ASD patients are more likely to experience gastrointestinal tract dysfunctions that include food allergies, dysbiosis, inflammatory bowel disease and indigestion [42,43]. These problems are connected to oxidative stress and may induce mitochondrial dysfunctions [44]. Maintaining the right balance of microbiota is vital for many processes in the organism. Numerous studies show the different composition of gut microbiota in ASD patients, but the direct consequences of this observation should be explored. The complexity of the problem and the heterogeneity of its aetiology in cases of autism necessitate the collection of further data on the importance of the gut–brain axis and its probable role in the pathophysiology of the condition.

The next significant factor that influences oxidative stress and contributes to the pathophysiology of ASD is mitochondrial dysfunction. Researchers have identified mitochondrial dysfunction in various types of ASD subjects, including those from ASD animal models and cell lines (e.g., lymphocytes and granulocytes) derived from children with ASD [45] to brain tissues of ASD patients [6,46,47]. Increasing research describes that there is an interaction between oxidative stress and mitochondrial function, which together affect the pathogenesis of ASD [28,36,46,48]. The mitochondrial electron transport chain is protected from damage caused by ROS by mitochondrial-specific superoxide dismutase and antioxidants such as glutathione (GSH). In ASD, the insufficiency of antioxidants together with inflammatory processes, environmental factors, DNA and mtDNA mutations influence mitochondrial dysfunction [6,46,48]. Mitochondrial dysfunction may be detected by elevated ROS markers. The dysfunction has an important impact on neuronal metabolism and astrocytes. The oxidative stress influences Na+/K+- ATPase, causing its malfunction that has multiple consequences on many metabolic pathways [49]. It is important to note that the mitochondrial electron transport chain is not only a source of free radicals but also a target of free radicals. Consequently, oxidative stress may damage mitochondrial function. Conversely, the abnormal mitochondrial function may cause further oxidative stress [28,36].

In addition to all of the above, there are also environmental factors, such as drugs and pollutant perinatal exposures [17], and other factors, such as high copper exposure or zinc deficiency, that can lead to an increase in oxidative stress in ASD [50].

The influence of oxidative stress on synaptic plasticity must be emphasised in the light of the above-mentioned metabolic changes associated with oxidative stress. Due to its high energy requirements, the brain consumes large amounts of oxygen, which leads to the production of ROS. Controlled ROS and RNS production provides the optimal redox state for the activation of transduction pathways involved in synaptic changes [51]. As has been described in autistic patients, the balance between antioxidant mechanisms and ROS production is disturbed. Therefore, oxidative stress acting in the early stages of life may negatively influence synaptic plasticity in ASD. This process is caused either by neuroinflammation, genetic mutations such as CAPRIN1 haploinsufficiency, mitochondrial dysfunction leading to neuronal tissue destruction, impaired calcium signalling and developmental/functional deficits of the neuronal network, including language deficits, attention deficit hyperactivity disorder and ASD [52].

Figure 1 presents an outline of the causes and consequences of oxidative stress in ASD.

All of the above mechanisms demonstrate the importance of oxidative stress in ASD. There is a great deal of interplay between the processes described, showing that many disorders can be both cause and consequence of oxidative stress. Considering the extent of these metabolic perturbations, it is clear that oxidative stress plays a role in the pathogenesis of ASD.

Specific biomarkers have been developed and evaluated to detect the presence of oxidative stress. Oxidative stress biomarkers are being investigated for their utility in ASD diagnosis.

## 3. The Role of Oxidative Stress as an ASD Biomarker

Although ASD is a recognised problem, as with many other neurodevelopmental and neuropsychiatric conditions, diagnosis is made based on clinical criteria defined by ICD-10 or the DSM V. To date, there is no universal biomarker for ASD. Finding biomarkers would allow for early diagnosis and shorten the time to start appropriate therapeutic intervention. Numerous studies have demonstrated oxidative stress in autism by finding its biomarkers. Nowadays, the usefulness of oxidative stress biomarkers in early diagnosis of ASD is being investigated [5].

There were many attempts to find the right biomarkers using blood, urine and cerebrospinal fluid samples. Direct ROS and RNS, such as superoxide, NO and peroxynitrite, are too reactive and/or have a short half-life, even shorter than 1 s. Therefore, such molecules cannot be isolated or measured directly in cells, tissues and body fluids. On the other hand, molecular products formed from the reaction of reactive oxygen and nitrogen species (RONS) with various biomolecules are generally more stable than RONS themselves. Thus, in most cases, RONS have commonly been traced by measuring their oxidation target products or antioxidants [5,53]. The oxidative stress biomarkers can be divided by their origin as protein-, lipid- or DNA-derived biomarkers and according to antioxidant type, such as enzymes and thiols. The most prominent substances in ASD are superoxide dismutase, catalase, glutathione peroxidase, malondialdehyde (MDA), ceruloplasmin and methionine [7,32].

Considering diagnostic utility and possible use in clinical practice, biomarkers can be divided into two categories: blood-based and urine-based.

The most recognised oxidative stress biomarker derived from blood is glutathione (GSH)—the most abundant non-protein thiol; it plays crucial roles in the antioxidant defence system and the maintenance of redox homeostasis in neurons [54]. Also, glutathione disulfide (GSSG), which is a product of glutathione oxidation, is considered one of the most pervasive markers in ASD patients, where its increased concentration is observed. Moreover, the glutathione-to-glutathione disulfide ratio GSH/GSSG, which was reduced in the plasma of ASD patients, is a significant biomarker indicating oxidative stress in these patients [34,54,55,56,57,58,59,60,61].

Between other prominent blood-derived oxidative stress biomarkers, MDA, SAH, nitric oxide and copper concentrations were significantly increased. In contrast, S-adenosyl *methionine,* S-adenosyl-L-*homocysteine* (SAM/SAH), methionine, cysteine, vitamins (B9, B12, D and E) and calcium concentrations were significantly reduced in children with ASD [62]. The next blood-based parameters potentially useful in ASD diagnosis are homocysteine (Hcy) and vitamin B6 [61]. Hcy is a non-protein amino acid derived from the methionine cycle required for activated methyl transfer and the trans-sulphuration pathway related to the synthesis of GSH [63]. Vitamins B6, B9 and B12 play important roles in the development, differentiation and functioning of the central nervous system. They are involved in the methionine–homocysteine pathway and, in that way, are connected with oxidative stress [64,65]. Together, homocysteine and vitamins B6, B9 and B12 are described as important substances in ASD pathophysiology because of the metabolic abnormalities in ASD patients, their gastrointestinal disorders and poor dietary habits [64]. However, assessing these parameters would be useful in detecting some additional disorders or nutritional deficiencies. The levels of these parameters, especially homocysteine, are too heterogeneous to be used as ASD biomarkers [5].

Other described blood-based biomarkers are isoprostanes, MDA and 4-hydroxynonenal, which result from lipid peroxidation, a widely described process in ASD. The use of these parameters in clinical practice should be considered, but more studies are necessary to introduce them as a diagnostic tool [9].

Further important biomarkers are 8-hydroxy-2′-deoxyguanosine and nuclear factor erythroid-2-related factor 2 (NRF2). NRF2 is a cytoprotective transcription factor that regulates the expression of genes responsible for encoding antioxidant, anti-inflammatory and detoxifying proteins and regulates cytoprotective genes [66,67]. The majority of studies have demonstrated an elevation in the serum concentration of NRF2 in patients diagnosed with ASD [68]. However, the data are not consistent.

8-hydroxy-2′-deoxyguanosine (8-OH-dg) is a product of oxidatively damaged DNA formed by hydroxy radicals, singlet oxygen and direct photodynamic actions [69]. 8-OH-dg is widely used as an oxidative stress biomarker; its elevation is described both in urine and plasma samples of ASD patients [70,71], but the data lack selectivity.

Because of the ease of urine collection, urine-based biomarkers have good potential for use in clinical practice. However, diagnostic methods have some limitations because urine production depends on renal function and may be altered by renal disease.

Most studies show the potential usefulness of blood and urine antioxidant capacity, antioxidant enzyme activity and redox intermediates. Other studies have found higher levels of homocysteine in the blood and urine of children with ASD. Hcy levels in urine and blood have been shown to be comparable in ASD patients, so urine measurement may be the preferred option [72,73], although more research is needed because of the variable results on the values of Hcy concentrations.

Other biomarkers that can be measured in urine are isoprostanes and their indirect markers like phospholipase A2 [70,74]. Isoprostanes are classified as reliable oxidative stress biomarkers. They are stable end products formed by the fermentation of hydroperoxides generated in lipid peroxidation [75].

In addition to biochemical parameters, oxidative stress causes loss of cell membrane structure, including changes in its fluidity and permeability, which can be measured and used as biomarkers [9,76,77].

In conclusion, many biomarkers of oxidative stress can be detected in patients with ASD. More data are needed to assess their selectivity and sensitivity to use their detection as a diagnostic tool for ASD [74]. In addition, there are insufficient data to correlate the described biomarkers with disease severity or as a response to potential therapies. Another issue is the diagnostic methods and their cost-effectiveness. Many authors emphasise that the use of complex measurement methods with metabolomic techniques would be a good choice for the future diagnosis of ASD but that more research is needed to introduce them into clinical practice.

In Table 1, the author summarises biomarkers potentially useful in ASD diagnosis.
biomedicines-13-00388-t001_Table 1Table 1Description of the oxidative stress biomarkers in ASD.Oxidative Stress BiomarkerMediumCharacteristic of the Biomarker in ASD Patients ReferencesGSSGBloodElevation[54,55,56,57,58,59,60,61]GSH/GSSGBloodReduction[54,55,56,57,58,59,60,61,77]glutathione peroxidaseBlood(plasma, whole RBC)Elevation/reduction—Inconclusive—more research needed[7,78]nitric oxide and its metabolites (nitrate and nitrite)Blood, urineElevation[3,79]superoxide dismutase SODBloodPlasmaRBCElevation/reduction—Inconclusive—more research needed[30,80,81,82]SAHBloodElevation[77]*S-adenosyl methionine*, S-adenosyl-L-*homocysteine*(SAM/SAH)BloodReduction[77]methionineBloodReduction[77,83]cysteineBloodReduction[77]homocysteineBlood/urineElevation[64,83,84]ceruloplasminBlood (serum)Elevation (inconclusive statistical difference)[64,82,83,84,85]copper concentrationsBloodElevation[50,64,82,83,84,85]calciumBloodReduction[86,87]malondialdehyde (MDA)UrineBloodElevation[62,80,88]4-hydroxynonenalUrine, plasma, RBC membranesElevation[70,74,89]lipoprotein-associated phospholipase A2UrineBlood (serum)Elevation[70,74,90]8-hydroxy-2′-deoxyguanosineUrineElevation[70,71]NRF2Blood (serum, monocytes)Reduction/elevation—Inconclusive—more research needed[68,91,92]vitamins (B9, B12, D and E)BloodPlasma, serumReduction[83,93,94,95]loss of cell membrane structureBloodChanges in its fluidity and permeability that can be measured[9,76,96]


## 4. The Role of Oxidative Stress as a Treatment Target

Nowadays, there is constant research of substances that could be used to cure autism, but there is no effective drug that can alleviate the core symptoms of ASD.

The available pharmacological treatments are prescribed to correct comorbid symptoms of ASD, like sleep disorders, aggressiveness and irritability, hyperactivity and attention deficit [97]. In the United States, the FDA (Food and Drug Agency) approved drugs for autism are risperidone and aripiprazole to help with irritability and aggression [98].

Considering the role of oxidative stress as a pathogenic factor, there is growing evidence of potential therapies based on antioxidant effects in alleviating ASD symptoms. Antioxidants potentially useful in ASD treatment are summarised in Table 2 below.

In their reviews, E. Zamberelli et al. and Bonomi et al. listed antioxidant substances potentially useful in the treatment of sleep disturbances in children with ASD: melatonin, tryptophan, coenzyme Q10, L-carnosine, luteolin and quercetin [97,99].

The study results showed that treatment with melatonin, with doses varying from 2 to 10 mg/day, was found to be well tolerated and effective in shortening sleep-onset latency, reducing the number of awakenings per night and bedtime resistance and increasing total sleep time [99]. That is consistent with numerous primary studies on melatonin [100,101]. Additionally, improvements during treatment with melatonin were observed in reducing externalising disruptive behaviours and parenting stress [102]. The authors underline the role of melatonin in multiple metabolic pathways, not only as a circadian rhythm regulator but also as an antioxidant and anti-inflammatory agent. Additional studies on animal models show the relationship between sleep deprivation and oxidative stress [103,104]. Melatonin lowers the formation of free radicals and protects ATP synthesis at the mitochondrial level [105]. These observations may explain its effectiveness in ASD patients [99,106].

The next well-described substance was tryptophan. As an essential amino acid and melatonin precursor, tryptophan was supplemented in the form of tryptophan-enriched cereals. In the actimetry study, patients were compared with those receiving “control cereals” without tryptophan. The studies revealed higher sleep efficiency, reduced sleep latency and better total activity levels in those receiving the enriched cereals [107].

The subsequent substance to be examined was L-carnosine, a non-enzymatic antioxidant that has been demonstrated to ameliorate cell energy metabolism, enhance immune response, regulate the metabolism of RNS and modulate the glutamatergic system. The study on the effect of treatment with L-carnosine revealed a statistically significant reduction in sleep disturbance during supplementation, as measured by sleep questionnaires and an autism severity scale [108,109].

The next well-described antioxidant with a potential role in the treatment of autism is coenzyme Q10 (ubiquinone and ubiquinol). Coenzyme Q10 is a lipid-soluble benzoquinone involved in oxidative phosphorylation as a cofactor for enzyme complexes in the mitochondrial membrane. It also has a recognised role as a free radical scavenger. Researchers showed improved sleep in patients receiving high doses (60 mg/day) of ubiquinone, as measured by the Childhood Autism Rating Scale. In addition, the authors observed some improvements in biochemical parameters of oxidative stress [110].

Another proposed group of antioxidants in ASD treatment were the flavonoids luteolin and quercetin. Both substances are Nrf2 activators and were administered in combination as dietary supplementation. The authors described their positive effect on behaviour. Both described substances have additional anti-inflammatory function [111], but this direct effect was not assessed in the studies. Moreover, transient irritability and problems with sleep were listed as side effects of that therapy. There are more randomised studies needed to assess the potential benefits of this kind of therapy.

In the other study on the role of antioxidants in the treatment of ASD, Castejon AM et al. evaluated the efficacy of cysteine-rich whey protein isolate (CRWP), a potent glutathione precursor, in addressing behavioural disturbances. This investigation was informed by the findings of previous studies that had demonstrated reduced glutathione concentrations in children with ASD. The 90-day supplementation regimen resulted in a substantial enhancement of glutathione levels and multiple domains of behaviour associated with ASD. The behavioural domains that demonstrated significant enhancements included socialisation, adaptive behaviour and internalising and maladaptive behaviour. However, the results obtained on behavioural scales were found to be comparable to those observed in the placebo group [112].

Given the importance of glutathione in oxidative stress, many authors underline the role of NAC (N-acetylcysteine) in the treatment of ASD. NAC is a synthetic derivative of the endogenous amino acid L-cysteine and a precursor of GSH [5,113,114]. The study of NAC treatment by Lee T et al. shows that this intervention can be an effective and well-tolerated option. Treatment outcomes were assessed using the Aberrant Behaviour Checklist and showed improvements in hyperactivity and irritability and improved social awareness in patients with ASD. The authors conclude that more research on NAC is needed before this treatment can be recommended [114]. Similar results were shown in the recent study by Ramkumar Aishworiya et al. In this study, the authors highlighted the important role of NAC in regulating the excitation/inhibition imbalance in ASD [115].

Another study of retrospective character, conducted by Cucinotta et al., gave preliminary results showing that so-called “metabolic support therapy” based on Q10 ubiquinol, vitamin E and complex-B vitamins brought favourable outcomes in patients with neurodevelopmental disorders [5,113,114]. The positive results were most pronounced in people with intellectual disability. The greatest improvements were seen in cognition, adaptive functioning and social motivation. The therapy was well tolerated with no serious adverse events [116,117].

Another treatment option may be antioxidant-rich foods, including broccoli, camel milk and dark chocolate, for ASD, but the results are difficult to standardise [5].

Asadabadi et al. presented a therapy based on the theory of the interplay between inflammation and oxidative stress. The clinical trial was based on a treatment consisting of risperidone in combination with celecoxib compared with risperidone plus a placebo. The results of the trial were assessed using the Aberrant Behaviour Checklist-Community Rating Scale and showed superiority of the use of risperidone with celecoxib in the treatment of irritability, social withdrawal and stereotypy [118].

The next group of compounds that have been investigated for their therapeutic potential in ASD are Nrf2 activators. Nrf2 is a transcription factor involved in immunological dysregulation/inflammation, oxidative stress and mitochondrial dysfunction. During oxidative stress, Nrf2 binds to specific DNA loci-antioxidant response elements (AREs). NRF2–ARE binding can regulate the expression of hundreds of cytoprotective genes, including antioxidant proteins and phase II enzymes [5,36]. NrF2 activators have anti-inflammatory and antioxidant effects. In this group, most of the studies were focused on sulforaphane, resveratrol, naringenin, curcumin and agmatine [119,120,121,122,123]. Yang J et al., in their review study on Nrf2 activators, underline that the results of preclinical and a few clinical studies are promising, but more randomised trials are needed to introduce this type of treatment [36].

As previously outlined, a substantial body of research has indicated the potential for antioxidant treatment in the management of ASD. However, the findings of these studies have revealed a divergence in the efficacy and durability of therapeutic outcomes, with the majority exhibiting transient positive results. This variability is likely attributable to individual genetic and environmental factors as well as small sample sizes and, sometimes, small ethnic diversity. The evaluation of research outcomes has predominantly employed behavioural scales, utilising structured questionnaires for caregivers or therapists, and occasionally presenting results in predetermined scales or through caregiver observations. This methodological approach may be subject to bias, as it is based on subjective opinions. Future research should aim to identify specific biomarkers that could facilitate the assessment process. The above-mentioned studies of antioxidants show promising results in improving sleep and some behavioural disturbances in children with ASD.

The present situation calls for multicentre randomised trials to collect comprehensive data and introduce antioxidant therapy into clinical practice.
biomedicines-13-00388-t002_Table 2Table 2Antioxidative substances potentially useful in ASD treatment.SubstanceMechanismEffectsReferencesMelatonincircadian rhythm regulator, mitochondria-targeted antioxidant and anti-inflammatory agentshortening sleep-onset latency, reducing the number of awakenings per night and bedtime resistance, and increasing total sleep time, minimalising disrupting behaviors, improving caregivers quality of life[100,102,124]Tryptophanessential amino acid and melatonin precursorhigher sleep efficiency, reduced sleep latency and better total activity[107]L-carnosinenon-enzymatic antioxidant, ameliorates cell energy metabolism,enhance immune response,regulate the metabolism of RNS, modulate the glutamatergic systemstatistically significant reduction in sleep disturbances[108,109]Q10 (Ubiquinone and Ubiquinol)cofactor for enzyme complexes in the mitochondrial membrane involved in oxidative phosporylation, a free radical scavengersleep improvement when using high doses (60 mg/day) of ubiquinone[116,117]Luteolin and quercetinNrf2 activator,suppress oxidative damage and lipid peroxidation and loss of antioxidant enzymes, including catalase and SOD, DNA-protective effect against H_2_O_2_, anti-inflammatory potentialpositive effects on behaviour;have side effect described as transient irritability[111,116,125]Cysteine-rich whey protein isolate (CRWP)a potent glutathione precursor that increases glutathione concentrationThe 90-day supplementation resulted in significantly improved socialisation, adaptive behaviour, and internalising and maladaptive behaviour, but overall results in behavioural scales were comparable to the placebo group[112]“metabolic support therapy” based on Q10 ubiquinol, vitamin E, and complex-B vitaminsenzymes cofactorsFavourable outcomes in cognition, adaptative functioning and social motivation. The therapy was well tolerated without any severe adverse events. Needs standardisation.[74,88,116]NAC (N-acetylcysteine)a synthetic derivative of the endogenous amino acid L-cysteine and a precursor of GSHglutamatergic modulatorImprovement in hyperactivity and irritability and enhanced social awareness in Aberrant Behavior Checklist Scale[113]Antioxidant-rich foods, including broccoli, camel milk and dark chocolatedepending on substanceSome improvements in behaviour, but the results are difficult to standardise[5,126,127]Risperidone in combination with celecoxibanti-inflammatoryReduction in irritability, social withdrawal and stereotypy[118]Sulforaphane, resveratrol, naringenin, curcumin, agmatineNrf2 activatorsImprovement of irritability and hyperactivity symptoms[36,118,121,126,128]


## 5. Discussion

The literature search results presented show the important role of oxidative stress in human organisms. It influences numerous metabolic pathways vital for energy metabolism, protein synthesis, defence mechanisms, DNA formation, transcription, neurotransmission and more.

The presence of oxidative stress in ASD is well proofed with the use of multiple biomarkers and functional studies. Taking into consideration the presence of this disturbance, there are continuous attempts to use that knowledge in the management of ASD patients.

The pathogenesis of ASD is known to be multifactorial; the role of oxidative stress, despite its influence on numerous metabolic pathways, must be taken into account. Data suggest that oxidative stress in ASD is a result of a combination of genetic and environmental factors, such as diet and inflammation. Conversely, oxidative stress influences metabolic pathways impacting genetic and inflammatory processes in a vicious circle. Still, it is impossible to determine if oxidative stress is a cause or symptom of ASD. However, growing knowledge of the role of oxidative stress is very important in further research and may be useful in ASD management.

There are attempts to use oxidative stress biomarkers in early ASD diagnosis. Finding easy-to-detect biomarkers would be very helpful in shortening the time for the right diagnosis and the delay in introducing therapeutic interventions and also minimising the costs of differential diagnosis. Many studies have found oxidative stress biomarkers potentially helpful, but data on the sensitivity and selectivity of that diagnostic tool are still lacking. Most researchers compared ASD patients’ results to those obtained from healthy controls. There can be bias when concerning patients with inflammatory disorders or energy deficits due to some genetic and/or metabolic disorder. A complex biomarker assessment in combination with behavioural scales would probably be useful. As mentioned in the section above, the clinical utility of a panel consisting of glutathione, the tGSH/GSSG ratio, isoprostanes, homocysteine and vitamin B can be the preferred option, but there is a need for further studies.

Finally, the utility of oxidative stress as a treatment target is another important field of research. There are many substances with antioxidant and anti-inflammatory functions with good safety profiles. The most promising results are described for melatonin, coenzyme Q10, flavonoids, L-carnosine and an antioxidant-rich diet. The results of the trials were mainly assessed using behavioural scales based on the parents’ and, sometimes, the investigator’s impressions. Some of the proposed antioxidant substances used in the research studies show side effects, such as increased irritability for luteolin and quercetin use.

There is a need for more objective measures to assess the effectiveness and durability of these interventions. The treated groups were heterogeneous in terms of age, ethnicity, diet prior to the intervention, comorbidity and time of intervention. In addition, the effectiveness of these interventions needs to be evaluated in more detail. Probably, because of the heterogeneity of ASD patients, including aetiology, dietary habits, comorbidities and kind and severity of symptoms, treatment interventions should be more individualised. During the antioxidant treatment, it is worth underlining that, to some extent, oxidative stress has an important regulatory role in organisms [9,129]. It is involved in processes such as cell signalling during apoptosis, cell proliferation, gene expression, defence mechanisms generated in phagocytic cells and others vital to maintaining cell homeostasis. This important role of oxidative stress should be considered during antioxidant treatment.

## 6. Conclusions

This paper attempts to summarise the current understanding of the role of oxidative stress in ASD. As described above, oxidative stress plays an important role in ASD pathogenesis through its direct influence on cells and its role in inflammation and the interplay with mitochondrial dysfunction and gene expression.

The growing body of data on oxidative stress and its role in autism is leading to attempts to translate this knowledge into clinical practice. Many researchers are trying to find oxidative biomarkers that may be useful in early diagnosis of ASD. However, data on their selectivity and sensitivity are still lacking. Many studies have used liquid chromatography–tandem mass spectrometry, gas chromatography–mass spectrometry and ELISA kits [5]. In the future, standardised and dedicated panels measuring multiple parameters using metabolomic techniques are likely to be used. Because of the young age and bioethical concerns, urine-based analysis may be the preferred option. More research is needed before it can be used as a diagnostic tool. Randomised multicentre trials are recommended to avoid small-group and ethnic bias. It would be advisable to assess patients as soon as possible after diagnosis or even to try to assess children from groups at increased risk of neurodevelopmental disorders.

Similarly, many attempts have been made to use oxidative stress as a treatment target in ASD, with positive results for melatonin, tryptophan, flavonoids, NAC, Nrf2 activators and some combined treatment and dietary approaches. These substances have shown efficacy in alleviating specific behavioural and sleep problems. It is recommended that individual nutritional status and specific problems are assessed to optimise the treatment strategy. However, more randomised, long-term trials using standardised medications are needed to assess their safety and efficacy before they can be introduced as a recommended treatment. The next step will be clinical trials with specific antioxidant drugs.

Oxidative stress plays an important role in many aspects of ASD pathophysiology but also as a potential diagnostic tool and treatment target. Further research is needed to translate these findings into clinical practice.

## Figures and Tables

**Figure 1 biomedicines-13-00388-f001:**
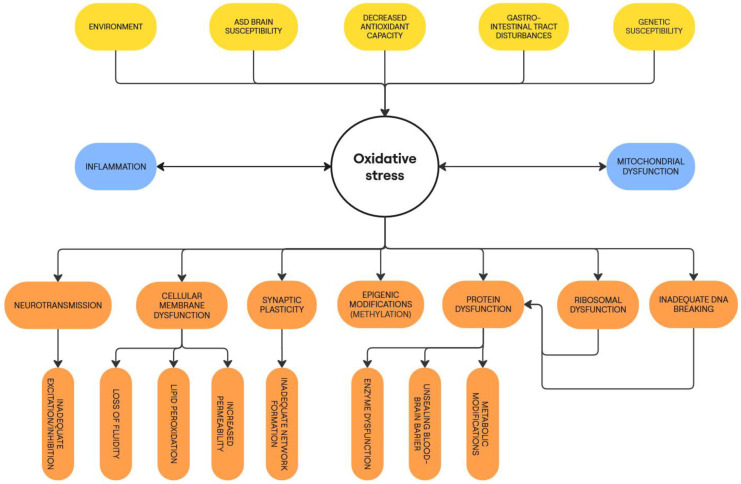
Causes and consequences of oxidative stress in ASD pathophysiology.

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
