# Peer review of "The Role of Oxidative Stress in Autism Spectrum Disorder Pathophysiology, Diagnosis and Treatment"

_biomedicines, 2025, doi:10.3390/biomedicines13020388_

Round 1

Reviewer 1 Report

Comments and Suggestions for Authors

Journal: Biomedicines (ISSN 2227-9059)

Manuscript ID: biomedicines-3426545

Type: Review

Title: The role of oxidative stress in autism spectrum disorder pathophysiology, diagnosis and treatment

Authors: Aleksandra Kuźniar-Pałka *

Section: Neurobiology and Clinical Neuroscience

Special Issue: Progress in Neurodevelopmental Disorders Research

General concept comments

The manuscript reviews the role of oxidative stress in Autism Spectrum Disorder (ASD), covering its pathophysiological implications, diagnostic potential, and treatment options. The organization is solid, though some sections are repetitive, and more concise language could be used. Highlighting actionable clinical recommendations would enhance its practical relevance, especially given the growing interest in oxidative stress as a biomarker and therapeutic target.

Specific comments

Abstract:

Avoid general statements like "data support the importance of oxidative stress but also a need for further research" without specifying gaps that require attention. However, consider tightening the language to improve clarity and impact. Suggest including specific data or results that highlight the study's significance.

-What are specific oxidative stress biomarkers that have good potential to be used as diagnostic tools supporting early ASD diagnosis?

Introduction:

The introduction thoroughly outlines the relevance of ASD and oxidative stress but might benefit from more recent prevalence statistics and additional context on why oxidative stress is a particularly relevant focus for ASD research.

-Line 40-41: “There is vast evidence that (4–13) has indicated the presence of oxidative stress in individuals with ASD patients and some also in their parents (9)”. The author should clarify and show specific evidence of oxidative stress-related ASD.

-Clarify the relationship between oxidative stress, mitochondrial dysfunction, and neuroinflammation early to set up the subsequent sections.

-Lines 44-48 need to be checked and clarified.

Some statistical information on ASD prevalence could be updated or cited for accuracy.

Materials and Methods:

The description of the literature search is brief and lacks specifics such as the inclusion/exclusion criteria, keywords used, or publication date range. Providing a diagram or summary table would improve transparency and reproducibility.

Results:

The results section is comprehensive, but key findings could be highlighted more effectively.

Tables and figures are helpful but could be visually more engaging with better legends and color schemes.

Discussion:

The discussion integrates findings well but could benefit from clearer segmentation into subsections, such as strengths, limitations, and future directions. The emphasis on the interplay between oxidative stress and inflammation is strong, but a more critical analysis of controversial findings is recommended.

Conclusion:

The conclusion needs to summarize the key points effectively and could provide more actionable recommendations for researchers and clinicians. Rewriting for this part is the recommendation.

Validity of the findings:

The findings are supported by current literature and well-referenced, but additional details on how biases were minimized in study selection would strengthen the validity.

Addressing potential limitations in the datasets reviewed (e.g., sample size, geographic variability) would be valuable.

Author Response

Abstract:

Avoid general statements like "data support the importance of oxidative stress but also a need for further research" without specifying gaps that require attention. However, consider tightening the language to improve clarity and impact. Suggest including specific data or results that highlight the study's significance.

-What are specific oxidative stress biomarkers that have good potential to be used as diagnostic tools supporting early ASD diagnosis?

- the  changes were applied to the article

Introduction:

The introduction thoroughly outlines the relevance of ASD and oxidative stress but might benefit from more recent prevalence statistics and additional context on why oxidative stress is a particularly relevant focus for ASD research.

-Line 40-41: “There is vast evidence that (4–13) has indicated the presence of oxidative stress in individuals with ASD patients and some also in their parents (9)”. The author should clarify and show specific evidence of oxidative stress-related ASD.

-Clarify the relationship between oxidative stress, mitochondrial dysfunction, and neuroinflammation early to set up the subsequent sections.

-Lines 44-48 need to be checked and clarified.

Some statistical information on ASD prevalence could be updated or cited for accuracy.

- the changes were applied to the article but probably need completion

Materials and Methods:

The description of the literature search is brief and lacks specifics such as the inclusion/exclusion criteria, keywords used, or publication date range. Providing a diagram or summary table would improve transparency and reproducibility- as it was narrative review on the suggestion of the Valentina Stoian the section was removed.

Results:

The results section is comprehensive, but key findings could be highlighted more effectively.

Tables and figures are helpful but could be visually more engaging with better legends and color schemes.

The author attempted to highlight the key findings.

Discussion:

The discussion integrates findings well but could benefit from clearer segmentation into subsections, such as strengths, limitations, and future directions. The emphasis on the interplay between oxidative stress and inflammation is strong, but a more critical analysis of controversial findings is recommended.

The author made some improvements - probably morf clarification would be useful.

Conclusion:

The conclusion needs to summarize the key points effectively and could provide more actionable recommendations for researchers and clinicians. Rewriting for this part is the recommendation.

Due to limited time author didn't managed to prepare recommendations.

Validity of the findings:

The findings are supported by current literature and well-referenced, but additional details on how biases were minimized in study selection would strengthen the validity.

Addressing potential limitations in the datasets reviewed (e.g., sample size, geographic variability) would be valuable.

Reviewer 2 Report

Comments and Suggestions for Authors

The article titled "The Role of Oxidative Stress in Autism Spectrum Disorder Pathophysiology, Diagnosis, and Treatment" by Aleksandra Kuźniar-Pałka offers valuable insights; however, there are several areas that require attention:

  1. The authors can revise the abstract to emphasize the background, aim of the study, methods, results, and conclusion.
  2. In the methodology section, the author may include details on specific search terms, years of the papers used, inclusion/exclusion criteria, data extraction, or methods employed to select relevant studies.
  3. Authors can create illustrations associated with antioxidant substances and their molecular mechanisms that potentially inhibit ASD.
  4. The author should discuss the challenges and limitations associated with using biomarkers in clinical practice.
  5. Authors should discuss and create a table outlining the underlying mechanisms related to biomarker expression based on in vitro and in vivo models of ASD.
  6. Authors may provide recommendations for future directions that focus on the long-term effects of antioxidant treatments in subgroups of individuals with ASD.
Comments on the Quality of English Language

The English could be improved to more clearly express the studies.

Author Response

  1. The authors can revise the abstract to emphasize the background, aim of the study, methods, results, and conclusion. The author attempts to revise this section of the article.
  2. In the methodology section, the author may include details on specific search terms, years of the papers used, inclusion/exclusion criteria, data extraction, or methods employed to select relevant studies- The author needs more time to clarify this section.
  3. Authors can create illustrations associated with antioxidant substances and their molecular mechanisms that potentially inhibit ASD. -The author needs more time to prepare illustration.
  4. The author should discuss the challenges and limitations associated with using biomarkers in clinical practice.- Some of these aspects were improved in the section but probably need more clarification.
  5. Authors should discuss and create a table outlining the underlying mechanisms related to biomarker expression based on in vitro and in vivo models of ASD.
  6. Authors may provide recommendations for future directions that focus on the long-term effects of antioxidant treatments in subgroups of individuals with ASD.- The author needs more time to prepare that more comprehensive recommendations.

Round 2

Reviewer 1 Report

Comments and Suggestions for Authors

The manuscript covers an essential topic with significant scientific and clinical implications.

Reviewer 2 Report

Comments and Suggestions for Authors

Accept in present form